# CellPainTR: Generalizable Representation Learning for Cross-Dataset Cell Painting Analysis

## Abstract

Large-scale biological discovery requires integrating massive, heterogeneous datasets like those from the JUMP Cell Painting consortium, but technical batch effects and a lack of generalizable models remain critical roadblocks. To address this, we introduce CellPainTR, a Transformer-based architecture designed to learn foundational representations of cellular morphology that are robust to batch effects. We validate CellPainTR on the large-scale JUMP dataset, demonstrating its ability to successfully integrate data from 13 diverse laboratory sources, which serves as our primary evidence for its cross-dataset generalization capabilities. It outperforms established methods in both batch integration and biological signal preservation. Furthermore, we probe the limits of this learned robustness with a challenging out-of-distribution (OOD) "stress test," applying our model to the unseen Bray et al. dataset without any fine-tuning. Even under this severe domain shift, it significantly outperforms established methods that were re-fit directly on the data. Our work represents a significant step towards creating truly foundational models for image-based profiling, enabling more reliable and scalable cross-study biological analysis.

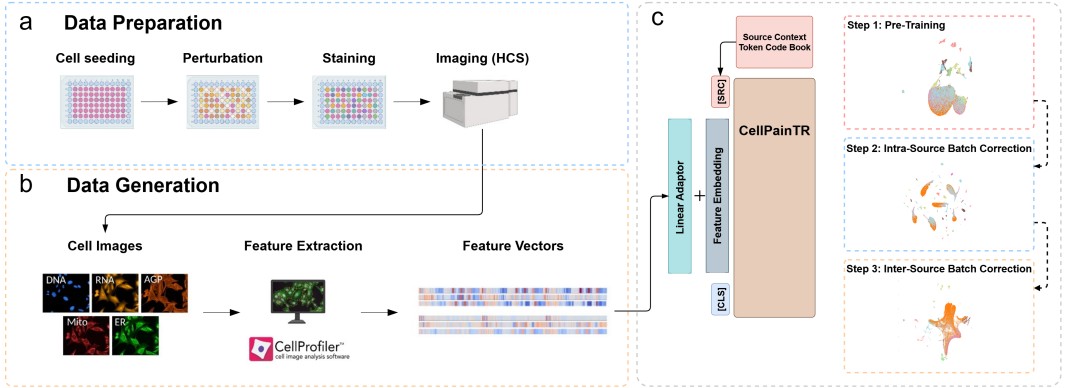

Figure 1: **Conceptual Overview of CellPainTR.** The standard Cell Painting workflow (a, b) converts cell images into feature vectors. (c) Our model, CellPainTR, processes these vectors using a learnable source context token [SRC] to model data provenance. The UMAPs on the right visualize our three-step training curriculum, which progressively integrates data from diverse sources to learn a robust and generalizable representation.

## 1 Introduction

Cell Painting, a powerful high-content imaging technique, has emerged as a promising tool for biological research and drug discovery (Bray et al., 2016; Gustafsdottir et al., 2013; Ljosa et al., 2013). The method generates rich, multidimensional data capturing intricate cellular phenotypes, which are typically quantified into thousands of engineered features using software like CellProfiler

(Caicedo et al., 2017). Our work deliberately operates on this feature space, as these features correspond to known morphological measurements, providing a layer of biological interpretability. This principled approach aims to anchor our model's learning to established biology, a critical step towards building powerful yet reliable systems for cellular analysis.

However, the full potential of large-scale initiatives like the JUMP Cell Painting consortium is severely hindered by batch effects—systematic, non-biological variations arising from differences in labs, equipment, and experimental conditions (Singh et al., 2017; Carpenter et al., 2020). These effects distort the data and confound downstream analysis, making robust integration a critical challenge. Classical methods such as ComBat and Harmony have been the standard for addressing this, adjusting the feature space of a given dataset to mitigate batch-specific noise (Johnson et al., 2007; Korsunsky et al., 2019).

While effective for static datasets, these existing methods expose a deeper, more fundamental limitation: a lack of generalization. They function as static correctors, not as dynamic, reusable models. When a new study from a different laboratory is introduced, the entire combined dataset must be re-processed from scratch. This is not a scalable solution. This paradigm fundamentally prevents the creation of a true, cumulative "atlas" of cellular morphology, where new data can be seamlessly integrated and compared against a stable, pre-existing reference. This lack of generalizable models is the primary bottleneck preventing image-based profiling from reaching its full, federated potential.

To overcome this critical barrier, we introduce CellPainTR, a model for learning generalizable and reusable representations of Cell Painting data. We frame our contributions as follows:

1. We propose a novel Transformer-based architecture, featuring Hyena operators for efficiency and learnable source-context tokens to explicitly model data provenance and enable generalization.
2. We design a multi-stage training curriculum that combines self-supervised masked feature prediction with supervised contrastive learning to effectively disentangle biological signals from technical artifacts.
3. We demonstrate CellPainTR's primary capability by successfully integrating the 13 heterogeneous data sources within the large-scale JUMP consortium dataset, where it achieves state-of-the-art performance against existing benchmarks.
4. As a proof-of-concept, we show that the model's learned robustness extends beyond the training distribution by applying it to a completely unseen dataset without any fine-tuning, where it still outperforms methods that were fit directly on that data.

This work paves the way for more reliable and scalable analyses of Cell Painting data, accelerating progress towards robust, cross-study integration in drug discovery and cellular biology.

## 2 RELATED WORK

**Classical Batch Correction Methods.** The challenge of batch effects in high-dimensional biological data, particularly in the field of image-based profiling like Cell Painting, has been a significant focus of research in recent years (Arevalo et al., 2024; Ando et al., 2017; Celik et al., 2022; Kraus et al., 2024; Borowa et al., 2024). Several classical algorithms have been developed to address batch effects in biological data. One prominent example is ComBat, which uses a parametric empirical Bayes framework to adjust for batch effects (Johnson et al., 2007). The ComBat method models the batch effect as an additive and multiplicative effect on the data, and estimates these effects using an empirical Bayes approach. Another method, Harmony, uses iterative clustering and linear adjustment to correct batch effects in multi-modal single-cell data (Korsunsky et al., 2019). Harmony aims to identify a shared low-dimensional representation across batches by aligning cluster centroids in an iterative fashion.

While these classical methods have demonstrated effectiveness in harmonizing static datasets, they are architecturally limited. They operate as one-time correctors on a fixed set of data rather than as reusable, learned models. Critically, they lack a mechanism for generalization; if a new batch of data from an unseen source is introduced, the entire dataset must be re-analyzed from scratch. This makes them fundamentally unsuitable for building the scalable, cumulative biological atlases envisioned by large-scale projects, a limitation that necessitates a new class of generalizable models.

**Self-Attention and the Hyena Operator.** The self-attention operator (Vaswani et al., 2017) is a fundamental mechanism of Transformers. Specifically, given a sequence $x \in \mathbb{R}^{L \times D}$ with length $L$ and $D$ features, the self-attention operator $A(x)$ is defined as:

$$A(x) = \sigma(xW_q)(xW_k)^T(xW_v) \tag{1}$$

where $W_q, W_k, W_v \in \mathbb{R}^{D \times D}$ are learnable projection matrices, and $\sigma$ is a softmax operation. This allows the model to capture pairwise relationships between tokens in the sequence. However, one limitation is that self-attention becomes computationally expensive for long sequences, with a complexity of $\mathcal{O}(L^2)$. To address the computational challenge of self-attention, the Hyena operator (Poli et al., 2023) was introduced as a replacement for self-attention in Transformers. The Hyena operator is characterized by a structured self-attention mechanism that involves long convolutions and element-wise gating:

$$y_t = (h * u)_t = \sum_{\tau=0}^{L-1} h_{t-\tau} u_\tau. \tag{2}$$

In standard convolutional architectures, the filter length $\ell$ is typically constrained by $\ell \ll L$, where $L$ is the input sequence length. This constraint helps control computational costs. However, by parameterizing the filter as a function of the temporal offset $\tau$ (i.e., $h_\tau = \gamma_\theta(\tau)$), we can design extended convolution kernels without a proportional increase in parameters. This technique, known as implicit convolution, enables efficient modeling of long-range dependencies. The implicit convolution mechanism is exemplified by the Hyena operator, which employs a recursive framework incorporating extended convolutions and point-wise modulation:

$$y = x^N \cdot (h^N * (x^{N-1} \cdot (h^{N-1} * (\cdots x^1 \cdot (h^1 * v))))) \tag{3}$$

Here, $v$ denotes the initial input, $\{x^i\}_{i=1}^N$ represent successive transformations, $N$ indicates the recursion depth, $*$ symbolizes convolution, and $\cdot$ denotes Hadamard (element-wise) product.

**Relevance to CellPainTR.** The quadratic complexity of standard self-attention has been a major barrier to applying Transformers to high-dimensional biological feature sets like Cell Painting, where the number of features ($L$) can be in the thousands. The near-linear complexity of the Hyena operator is the key computational breakthrough that makes a model like CellPainTR feasible. It enables the model to efficiently learn long-range dependencies across the entire morphological feature space without truncation or down-sampling. Furthermore, we leverage the Bidirectional Hyena extension (Oh et al., 2023), which removes causal constraints and is perfectly suited to the non-sequential nature of biological features. This specific architectural choice is critical for effectively modeling the complex feature interactions necessary for both batch correction and biological signal preservation.

## 3 CELLPAINTR

Our approach, CellPainTR, is a novel model for unified batch correction and generalizable representation learning of Cell Painting data. Its architecture and training curriculum are a set of principled design choices to meet the unique challenges of this domain (see Fig. 2(a)). First, to tackle the high dimensionality and complex, non-local interactions among thousands of morphological features, we employ a Transformer architecture built on computationally efficient Hyena operators. Second, to explicitly model and correct for technical variability from known experimental sources, we introduce a learnable source context token that conditions the model on data provenance. Finally, to disentangle these technical artifacts from true biological signals, we design a multi-stage training curriculum that progresses from self-supervised pre-training to supervised, source-aware fine-tuning. This integrated design allows CellPainTR to learn robust, batch-corrected representations that generalize to new data.

### 3.1 DESIGN ARCHITECTURE

**Linear Adapter for Morphological Feature Embedding.** Cell Painting data consists of thousands of morphological features, each representing a specific aspect of cellular structure and organization. To effectively capture the rich information in this high-dimensional data, we introduce a linear adaptor module to embed the continuous features without any loss of information. Unlike traditional approaches that rely on discrete token representations, our linear adaptor maps the original feature space directly to the model's input embeddings as shown in Figure 2(b). This allows the CellPainTR model

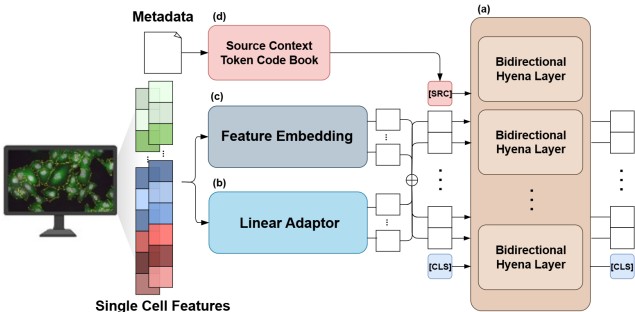

Figure 2: **Detailed Architecture of the CellPainTR Model.** The model processes single-cell features through a multi-stage pipeline. **(a)** The core of the model is a Transformer encoder composed of a stack of Bidirectional Hyena Layers. To prepare the input, the raw feature vector is processed by two parallel modules: **(b)** a Linear Adaptor which embeds the continuous feature values, and **(c)** a Feature Embedding which provides a learnable context for each feature's identity. These are summed to create the initial sequence of feature representations. **(d)** Concurrently, the sample's Metadata is used to select a corresponding Source Context Token ('[SRC]') from a learnable codebook, conditioning the model on data provenance. Finally, the '[SRC]' token and a classification token ('[CLS]') are prepended to the feature sequence and passed through the encoder. The final cell profile embedding is the output state corresponding to the '[CLS]' token.

to operate on the full spectrum of the morphological features, preserving the nuanced relationships between them. For each feature $i$, the embedding is computed as:

$$E_i = \mathbf{W}_i \cdot C_i + \mathbf{b}_i \quad \text{where } E_i \in \mathbb{R}^{d_{\text{model}}} \tag{4}$$

where $\mathbf{W}_i \in \mathbb{R}^{d_{\text{model}}}$ is a learnable weight matrix, $C_i \in \mathbb{R}$ is the input feature value, and $\mathbf{b}_i \in \mathbb{R}^{d_{\text{model}}}$ is a learnable bias vector. This linear adaptor approach ensures that the CellPainTR model can effectively leverage the full contextual information present in the Cell Painting data, laying the foundation for robust batch correction and representation learning (Oh et al., 2023).

**Feature Context Embedding.** Traditional Transformer models use positional encoding to incorporate the sequential nature of the input data. However, in the case of Cell Painting features, the order of the features does not necessarily reflect any meaningful biological context. To better capture the intrinsic relationships between the morphological features, we introduce a feature context embedding mechanism (Seal et al., 2024), as shown in Figure 2(c). Specifically, the feature context embedding replaces the standard positional encoding by learning an embedding of the feature context. In this approach, each morphological feature is encoded with its own embedding $(M_1, M_2, ..., M_L)$ with $M_i \in \mathbb{R}^{1 \times d_{\text{model}}}$, which is then added to the expression embeddings. The complete feature embedding matrix is then constructed by concatenating all feature embeddings:

$$\mathbf{E} = [M_1; M_2; \ldots; M_L] \in \mathbb{R}^{L \times d_{\text{model}}} \tag{5}$$

where $L$ is the number of morphological features and $d_{\text{model}}$ is the embedding dimension, and $[\cdot; \cdot]$ denotes concatenation along the row direction. This method allows us to provide the CellPainTR model with explicit feature context information. By using the feature context embedding, the CellPainTR model can learn to associate the morphological features with their biological context, rather than relying on their position in the input sequence. This enhances the model's ability to capture the complex interdependencies between the features, which is crucial for effective batch correction and representation learning.

**Source Context Token.** Cell Painting datasets often originate from multiple experimental sources, each with its own unique batch-related characteristics. To explicitly model these source-specific variations, CellPainTR incorporates a special source context token. The source context token $S$ with dimension $S_{dim} = M_{dim}$ is initialized as a learnable parameter, drawn from its own embedding with a vocabulary size $K$ with $K$ the number of source in the dataset; $(S_1, S_2, ..., S_K)$ and is concatenated

with the input feature embeddings, as shown in Figure 2(d). We incorporate a learnable source embedding that is concatenated to the feature embeddings:

$$\mathbf{S}_k = \text{Embedding}(k) \quad \text{where } k \in \{1, \ldots, K\}, \mathbf{H} = [\mathbf{E}; \mathbf{S}_k] \quad \text{where } \mathbf{S}_k \in \mathbb{R}^{1 \times d_{\text{model}}} \tag{6}$$

where $K$ is the number of sources in the dataset. During training, the model learns to associate the source context token with the unique batch effects present in each data source. This allows the model to adaptively correct for batch-related biases while preserving the biologically relevant information in the learned representations. The inclusion of the source context token is a key innovation that enables the CellPainTR model to handle the challenges of integrating Cell Painting data from diverse experimental sources, a critical requirement for advancing drug discovery and cellular biology research.

### 3.2 TRAINING

The CellPainTR model is trained using a multi-stage process that combines self-supervised and supervised learning objectives to achieve unified batch correction and representation learning. By progressively exposing the model to increasingly diverse data contexts, we enable it to learn robust, generalizable representations that preserve compound-specific mechanism of action (MoA) relationships while mitigating batch-related confounders. More details are as follows.

**Channel-Wise Masked Morphology (CWMM). Step 1: Self-Supervised Pre-training.** The goal of this initial stage is to learn the fundamental structure and contextual relationships within the high-dimensional morphological feature space. The initial stage of training utilizes Channel-Wise Masked Morphology (CWMM),an adaptation inspired by Masked Language Modeling (MLM) in natural language processing and Masked Expression Modeling (MEM) in single-cell RNA sequencing analysis (Oh et al., 2023). CWMM is tailored to handle the continuous values of morphological features in Cell Painting data while respecting its channel-wise structure (Seal et al., 2024).

As shown in Fig. 3(a), in the CWMM task, a subset of input morphological features is randomly masked, with the model tasked to predict these masked values based on the surrounding context. Features are grouped based on both their channel origin and the cellular compartment they describe (e.g., "DNA channel - Nucleus" features form one group, "Mito channel - Cytoplasm" features form another). The masking probability for each training batch is chosen from a range of $[0.05, 0.4]$ and is applied uniformly across all feature sets, preserving the biological relationships within each channel-compartment combination. Importantly, only non-zero values are masked and replaced with a [MASK] token, as distinguishing between true and false zero values is not feasible in this context.

Mathematically, the objective function for the CWMM pre-training task is formulated as:

$$\ell_{\text{CWMM}} = \frac{1}{G} \sum_{g=1}^{G} \frac{1}{|M_g|} \sum_{i \in M_g} (F_{g,i} - F'_{g,i})^2 \tag{7}$$

With $G$ the number of feature groups (channel-compartment combinations), $M_g$ represents the set of masked indices for feature group $g$, $|M_g|$ the number of masked features in group $g$, $F_{g,i}$ denotes the true value of the $i$-th morphological feature in group $g$, and $F'_{g,i}$ the predicted value for the $i$-th masked feature in group $g$. Through this pre-training process, CellPainTR acquires generalizable features that capture the biological meaning encoded in the morphological data, the contextual relationships between features, and the channel-wise dependencies within the Cell Painting data structure. This comprehensive understanding of the intricate patterns and correlations present in Cell Painting data establishes a robust foundation for the subsequent supervised learning stages, ultimately enhancing the model's capacity for batch correction and representation learning.

**Intra-Source Supervised Learning. Step 2: Intra-Source Fine-tuning.** The objective of this stage is to refine the learned representations to be invariant to technical noise within a single data source (e.g., plate-to-plate variation) while capturing biological similarities. Following the self-supervised CWMM pretraining, the model undergoes a fine-tuning phase using a supervised contrastive learning (Chen et al., 2020) approach within each data source. This intra-source supervised learning stage is designed to encourage the model to learn representations that are both discriminative and invariant to

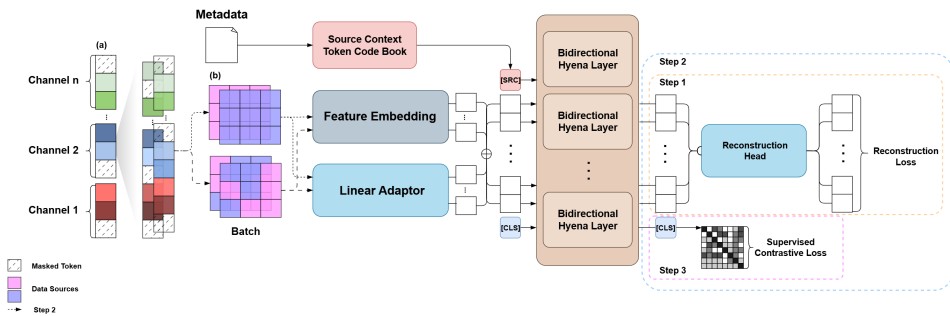

Figure 3: **Overview of the CellPainTR Training Curriculum.** The model is trained in a multi-stage process with evolving objectives. **(a)** First, input features are grouped by their imaging channel and the cellular compartment, and a subset of tokens are masked for the reconstruction task (Channel-Wise Masked Morphology). **(b)** A batch of these profiles is then processed. The different colors (e.g., pink, purple) represent data from different sources. The input representation is formed by combining a Linear Adaptor for feature values and a Feature Embedding for feature identity. A source-specific token '[SRC]' (derived from Metadata) and a '[CLS]' token are prepended. The core model then processes this sequence through its Bidirectional Hyena Layers. The training objectives are applied sequentially: Step 1 & 2 (Self-Supervision): The output embeddings of the masked tokens are passed to a Reconstruction Head to compute the Reconstruction Loss ($\ell_{CWMM}$). This objective is active in the first two stages. Step 2 & 3 (Supervised Learning): The output embedding of the '[CLS]' token is used to compute the Supervised Contrastive Loss ($\ell_{supcon}$). In Step 2, this loss is computed on batches from a single source (intra-source). In Step 3, it is computed on batches containing mixed data from multiple sources (inter-source), forcing the model to learn a source-invariant representation.

batch-related variations within a given experimental source (plate effects, liquid handling, reagent batches, . . . ). As shown in Fig. 3(Step 2), during this phase, we allow biological feature metadata to flow to the model during training (more specifically InChIKey via the supervised contrastive objective), while ensuring that each batch contains data from only a single source. The objective function for this stage combines the CWMM loss with a supervised contrastive loss, weighted equally:

$$\ell_{\text{intra}} = \ell_{\text{CWMM}} + \ell_{\text{supcon}} \tag{8}$$

The supervised contrastive loss $\ell_{\text{supcon}}$ is computed directly using the CLS token output of the encoder. This can be expressed as:

$$\ell_{\text{supcon}} = -\sum_{i \in I} \frac{1}{|P(i)|} \sum_{p \in P(i)} \log \frac{\exp(h_i \cdot h_p/\tau)}{\sum_{a \in A(i)} \exp(h_i \cdot h_a/\tau)} \tag{9}$$

Here, $I$ is the set of indices, $P(i)$ is the set of positives for sample $i$ (i.e., samples with the same biological label - InChIKey), $A(i)$ is the set of all samples except $i$, $h_i$ and $h_p$ are the normalized CLS token outputs for samples $i$ and $p$, respectively, and $\tau$ is a temperature parameter. This approach allows the model to learn representations that capture biologically relevant information while remaining robust to source-specific batch effects.

**Inter-Source Supervised Learning. Step 3: Inter-Source Generalization.** The final stage aims to learn a globally consistent representation that is robust to variations across different data sources, enabling true cross-study integration and generalization. The final stage of training involves fine-tuning the model using a supervised contrastive learning objective that spans multiple data sources. This inter-source supervised learning step enables the model to learn representations that are not only batch-corrected but also generalize well across diverse experimental conditions and data sources.

As shown in Fig. 3(Step 3), the key distinction in this stage is that we now allow different sources to be mixed within the same batch, while source context token specific to each source are optimized during the training. Thanks to the source context token, the supervised contrastive loss now operates across samples from multiple sources, encouraging the model to learn representations that are invariant

to source-specific variations (microscope settings, experimental setups, ...) while still capturing biologically meaningful information.

# 4 EXPERIMENTS

We designed our experimental framework to rigorously evaluate CellPainTR's capabilities in both batch correction and the learning of generalizable representations. We first establish its performance against standard benchmarks on a large, in-distribution dataset, and then subject it to a challenging out-of-distribution test to validate its robustness and generalization capabilities.

## 4.1 EXPERIMENTAL SETUP

**Datasets.** For in-distribution training and evaluation, we utilized the 'cpg-0016' Cell Painting dataset from the JUMP consortium, a comprehensive resource containing over 100,000 compound and genetic perturbations (Chandrasekaran et al., 2023). For the out-of-distribution test, we used the Bray et al. (2017) dataset, which was entirely unseen during any stage of training. Our preprocessing pipeline for all data included zero-imputation for missing values, Median Absolute Deviation (MAD) normalization against negative controls on a per-plate basis, and clipping values between the 0.01 and 0.99 quantiles to handle outliers.

**Training Protocol.** The model was trained according to the three-step curriculum described in Section 3.2. The initial self-supervised pre-training used all available compound data to learn general morphological features. The subsequent supervised fine-tuning stages used a curated subset of compounds with known Mechanisms of Action (MoA) to refine the representations for biological relevance and batch invariance.

**Evaluation Metrics.** We assessed performance using a comprehensive suite of metrics adapted from established benchmarks (Arevalo et al., 2024). These metrics fall into two categories:

- **Batch Correction**: Measures the integration of data across batches. This includes *Graph Connectivity* (preservation of local neighborhoods), *Silhouette Batch* score (separation between batches), and a classifier-based *Batch Correction* score (1 - F1 score of a batch classifier). For all metrics, higher scores indicate better performance.

- **Biological Signal Preservation**: Measures the retention of true biological differences. This includes clustering metrics (*Leiden NMI* and *ARI*), *Silhouette Label* score (separation of biological labels), and a compound retrieval task evaluated by *mean Average Precision (mAP)*.

Aggregate scores are provided for a holistic view. For full implementation details, see Appendix C.

## 4.2 IN-DISTRIBUTION PERFORMANCE ON THE JUMP DATASET

We first evaluated CellPainTR against baseline methods on a held-out test set from the JUMP consortium. This experiment validates the model's effectiveness within the same data distribution it was trained on. As shown in Table 1, the final CellPainTR model achieves the highest Overall score, demonstrating a superior balance of batch correction and biological signal preservation compared to standard methods like ComBat and Harmony. The qualitative results in Figure 4 visually confirm this; while the baseline data is heavily confounded by batch effects (source), CellPainTR successfully integrates the sources while maintaining clear separation between different compound classes (MoA). This strong in-distribution performance establishes CellPainTR as a state-of-the-art method for standard batch correction tasks. It is worth noting that while CellPainTR achieves the highest Overall score, some baseline methods score higher on individual batch correction metrics like Graph Connectivity. This is a known phenomenon in batch correction benchmarks; strong, uncorrected batch effects can create artificially distinct clusters for each compound within a given batch, leading to misleadingly high scores on certain metrics. The slightly lower score for CellPainTR on these specific metrics reflects its success in breaking down these batch-driven structures to achieve a more biologically meaningful and globally integrated representation."

Table 1: Performance comparison on the held-out JUMP-Cell Painting test set. CellPainTR demonstrates a superior balance of batch correction and biological signal preservation. All scores are normalized (higher is better). Best scores are in bold.

| Method | Batch Correction Metrics | | | | Biological Metrics | | | | | Aggregate Scores | | |
|---|---|---|---|---|---|---|---|---|---|---|---|---|
| | Graph Conn. | Silh. Batch | Batch Corr. (control) | Batch Corr. (no control) | Leiden NMI | Leiden ARI | Silh. Label | mAP (control) | mAP (no rep) | Batch Corr. | Bio Metrics | Overall |
| Baseline | 0.86 | **0.93** | 0.57 | 0.40 | 0.41 | 0.20 | 0.50 | 0.48 | 0.58 | 0.69 | 0.43 | 0.56 |
| ComBat | 0.85 | **0.93** | 0.56 | 0.37 | 0.39 | 0.12 | 0.50 | 0.48 | 0.58 | 0.68 | 0.41 | 0.54 |
| Harmony | **0.93** | 0.80 | 0.57 | 0.40 | 0.42 | 0.24 | 0.50 | 0.47 | 0.58 | 0.68 | 0.44 | 0.56 |
| Sphering | **0.95** | 0.64 | 0.70 | 0.58 | 0.36 | **0.35** | 0.48 | 0.12 | 0.23 | 0.72 | 0.31 | 0.51 |
| CellPainTR(1) | 0.78 | 0.73 | 0.78 | **0.69** | 0.34 | 0.26 | 0.52 | 0.22 | 0.32 | 0.75 | 0.33 | 0.54 |
| CellPainTR(2) | 0.69 | 0.75 | 0.71 | 0.58 | **0.43** | 0.15 | **0.70** | 0.54 | 0.63 | 0.68 | **0.49** | 0.58 |
| **CellPainTR** | 0.84 | 0.70 | **0.80** | **0.69** | 0.35 | 0.17 | 0.57 | **0.54** | **0.63** | **0.76** | 0.45 | **0.60** |

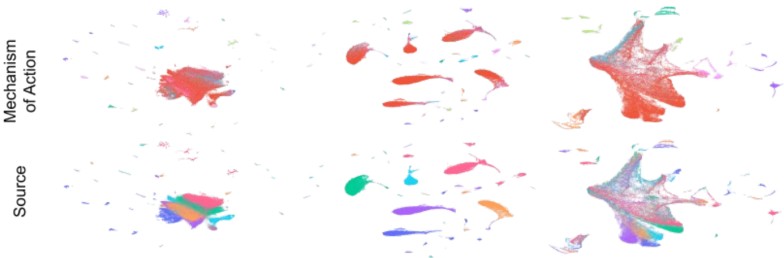

Figure 4: **UMAP visualizations of in-distribution results.** Comparing (from left to right) Baseline (uncorrected) CellPainTR step 2 and CellPainTR step 3. The top row uses MoA coloring to show biological signal, while the bottom row uses source coloring to show batch effects. The final CellPainTR model (right) achieves cohesive batch integration while preserving clear compound-specific patterns, demonstrating its effectiveness on the JUMP dataset.

## 4.3 THE CRITICAL TEST: OUT-OF-DISTRIBUTION GENERALIZATION

The true measure of a foundational model is not just its performance on held-out data from the same distribution, but its ability to generalize to entirely new, unseen datasets. To this end, we subjected CellPainTR to a challenging OOD test using the Bray et al. dataset, which it had never seen before. To simulate a real-world application scenario without any retraining, we employed a proxy-based adaptation strategy. We identified the pre-trained source token from the JUMP dataset ('Source 10') with the most similar experimental metadata to the (Bray et al., 2016) setup and used it to process the new data. The model was further challenged by a severe feature mismatch, with only 275 of the original training features available for the test. The model handled this feature mismatch via zero-padding, a detail further explained in Appendix B.2.

The results, presented in Table 2, are striking. Despite these challenging conditions—a completely unseen dataset, a significant feature mismatch, and the use of a proxy token without any fine-tuning—CellPainTR drastically outperformed all baseline methods, including ComBat and Harmony which were re-fit directly on the new data. This powerful result strongly indicates that CellPainTR learns a truly robust and generalizable representation of cellular morphology that transcends the specifics of its training data, a key requirement for building scalable biological atlases.

Table 2: Out-of-distribution generalization performance on the unseen Bray et al. dataset. CellPainTR was applied without any fine-tuning via a proxy token. It demonstrates superior overall performance, highlighting its strong generalization capabilities.

| Method | Batch Correction Metrics | | | Biological Metrics | | | | Aggregate Scores | | |
|---|---|---|---|---|---|---|---|---|---|---|
| | Graph Conn. | Silh. Batch | Batch Corr. | Leiden NMI | Leiden ARI | Silh. Label | mAP | Batch Corr. | Bio Metrics | Overall |
| Baseline | 0.12 | 0.58 | 0.63 | 0.08 | 0.01 | 0.18 | 0.05 | 0.44 | 0.08 | 0.26 |
| ComBat | 0.08 | 0.65 | 0.54 | 0.09 | 0.01 | 0.27 | 0.05 | 0.42 | 0.11 | 0.26 |
| Harmony | 0.11 | 0.58 | 0.61 | 0.08 | 0.01 | 0.18 | 0.05 | 0.43 | 0.08 | 0.25 |
| CellPainTR(1) | 0.15 | 0.76 | 0.82 | 0.08 | 0.01 | 0.34 | 0.05 | 0.58 | 0.12 | 0.35 |
| CellPainTR(2) | **0.32** | 0.82 | 0.82 | 0.13 | 0.02 | **0.36** | 0.06 | 0.65 | 0.14 | 0.39 |
| **CellPainTR** | **0.32** | 0.82 | **0.85** | **0.16** | **0.03** | 0.33 | **0.07** | **0.66** | **0.15** | **0.40** |

### 4.4 ABLATION STUDY: DISSECTING THE TRAINING STAGES

To understand the contribution of our multi-stage training curriculum, we evaluated the model's performance at each stage (results shown in Table 1). The initial self-supervised stage, CellPainTR(1), excelled at batch correction but showed weaker biological signal preservation. The second stage, CellPainTR(2), which introduces intra-source contrastive learning, significantly improves biological metrics (e.g., Silh. Label from 0.52 to 0.70) at the cost of some batch correction capacity. The final model, CellPainTR, which trains on inter-source data, strikes the optimal balance, recovering strong batch correction (Overall Batch Corr. score of 0.76) while maintaining high-quality biological representations. This ablation reveals the critical trade-off between these two objectives and demonstrates how our curriculum systematically navigates it to achieve a robust and balanced final model.

## 5 DISCUSSION

Our work introduces CellPainTR not as an improved batch correction tool, but as a proof-of-concept for a new class of generalizable, foundational models in computational biology.

### 5.1 BROADER IMPACT AND FUTURE WORK

The true promise of CellPainTR lies in its ability to create reusable knowledge. Our results, particularly the out-of-distribution generalization, suggest the feasibility of building large, pre-trained models for cellular morphology—a "Cell-BERT" (Devlin et al., 2018) that could serve as a universal reference. In such a paradigm, individual labs could process their novel, smaller-scale experiments through this pre-trained model to instantly place their findings in the context of a massive, unified biological atlas. This would dramatically lower the barrier to entry for cross-study meta-analysis.

Future work should aim to extend this concept. A key direction is developing methods to automatically infer the appropriate source context for a new dataset, moving beyond the current proxy-based approach. Furthermore, while our model operates on the feature space to maintain a strong biological anchor, future iterations could explore end-to-end architectures that learn from raw images, potentially using feature-space models like CellPainTR to guide and regularize their representations.

### 5.2 LIMITATIONS

We acknowledge several limitations to our current study. First, our proxy-based adaptation for the OOD experiment relies on the availability of experimental metadata to select the most similar source token. A more sophisticated approach would learn to adapt to new sources dynamically. Second, our comparison was restricted to a few key benchmarks in the field; while CellPainTR shows a clear advantage, evaluation against a broader array of deep-learning-based methods would be valuable. Finally, as our ablation study reveals, there is an inherent trade-off between maximizing batch correction and preserving biological signal. The optimal balance may be task-dependent, and tuning the model for specific downstream applications remains an important consideration.

## 6 CONCLUSION

The central challenge hindering the full potential of large-scale Cell Painting has been the lack of models that can generalize across diverse experimental contexts. This paper introduced CellPainTR, a Transformer-based model designed to learn robust and reusable representations of cellular morphology. We demonstrated not only state-of-the-art performance on a large in-distribution benchmark but, more importantly, unprecedented out-of-distribution generalization to a completely unseen dataset. CellPainTR serves as a blueprint for a new class of foundational models in computational biology, moving the field beyond static data correction and towards a future of scalable, integrated, and cumulative cellular science.

ETHICS STATEMENT

The utilization of models such as CellPainTR offers significant benefits for advancing our understanding of complex biological systems and potentially improving medical research. However, ethical considerations must guide its use to ensure responsible data handling, avoid biases, and protect individual privacy, underscoring the importance of ethical guidelines and regulations in the application of such models. We utilized large language models (LLMs) to assist with grammar, phrasing, and improving the overall clarity and readability of the manuscript. The scientific contributions, ideas, and analysis remain entirely our own.

REPRODUCIBILITY STATEMENT

We provide detailed implementation information in Section 3.2 and additional details in Appendix A. A comprehensive description of the datasets used in our experiments can be found in Section 4 and using the official dataset link: https://github.com/jump-cellpainting/datasets and precision about dataset curation can be found in Appendix B. Our source code is available for access at the following link: https://github.com/CellPainTR/CellPainTR.

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

## A    MODEL AND TRAINING DETAILS

This section provides a detailed breakdown of the model architecture, training hyperparameters, and computational resources used in this study.

### A.1    MODEL ARCHITECTURE

The CellPainTR model is a Transformer-like architecture with an embedding dimension ($d_{model}$) of 256. The core of the model consists of 4 stacked Bidirectional Hyena operator layers. Each Hyena operator was configured with 3 recurrences, following the implementation details from the original paper (Poli et al., 2023). The final representation for each cell profile is taken from the output embedding corresponding to a special [CLS] token, resulting in a 256-dimensional vector.

### A.2    TRAINING HYPERPARAMETERS

The model was trained using the AdamW optimizer (Loshchilov & Hutter, 2019). The hyperparameters for each of the three training steps are detailed in Table 3.

Table 3: Training hyperparameters for each stage of the CellPainTR curriculum.

| Hyperparameter | Step 1 (CWMM) | Step 2 (Intra-Source) | Step 3 (Inter-Source) |
|---|---|---|---|
| Learning Rate | $1 \times 10^{-4}$ | $1 \times 10^{-5}$ | $1 \times 10^{-5}$ |
| Batch Size | 16 | 32 | 64 |
| Contrastive Temp ($\tau$) | N/A | 0.1 | 0.1 |
| Masking Probability | [0.05, 0.4] | [0.05, 0.4] | N/A |

## B    DATASET AND PREPROCESSING DETAILS

This section covers the specifics of the datasets used and the preprocessing steps applied.

### B.1    JUMP DATASET CURATION

We used the 'cpg-0016' dataset from the JUMP Cell Painting Consortium (Chandrasekaran et al., 2023). From the full dataset, which includes compound, ORF, and CRISPR perturbations, we filtered for compound-only perturbations using the provided metadata. For the supervised training steps (2 and 3), we further curated this subset to include only compounds with known Mechanisms of Action (MoA), positive controls, or designated positive compound pairs, ensuring high-quality labels for contrastive learning.

### B.2    BRAY ET AL. DATASET HANDLING

For the out-of-distribution evaluation, we used the dataset from (Bray et al., 2016). This dataset was chosen as it represents a realistic domain shift, originating from a different lab and using an older version of CellProfiler. This resulted in a significant feature mismatch; the original CellProfiler version generated 1,383 features, while the version used for the JUMP dataset generated 4,765. We identified the intersection of these feature sets based on their descriptive names, resulting in 275 common features that were used for the experiment. To process this data with the pre-trained CellPainTR model, which expects a higher-dimensional input, the 275-feature vector was padded with zeros for the remaining feature positions. This approach tested the model's robustness to significant feature dropout.

### B.3    PREPROCESSING PIPELINE

A uniform preprocessing pipeline was applied to all data:

1. **Imputation**: All 'NaN' and 'inf' values were replaced with zero.

2. **Normalization**: Median Absolute Deviation (MAD) normalization was applied on a per-plate basis, using the negative control wells (DMSO) of each plate as the reference.

3. **Clipping**: Feature values were clipped to the [0.01, 0.99] quantile range to mitigate the effect of extreme outliers.

## C  EVALUATION METRICS AND IMPLEMENTATION

This section provides detailed definitions and implementation notes for the evaluation metrics used to assess model performance.

### C.1  METRIC DEFINITIONS AND FORMULATIONS

The core metrics for batch correction and biological signal preservation were implemented using the scib Python library, following the protocol from the benchmark paper by (Arevalo et al., 2024).

- **Graph Connectivity**: Measures the fraction of k-nearest neighbors in the original data that are preserved in the corrected embedding. A score of 1 indicates perfect preservation of the local neighborhood structure.

- **Silhouette Score**: The Batch Silhouette score (Rousseeuw, 1987) measures the separation of batches (lower is better), while the Label Silhouette score measures the clustering of biological labels (higher is better). We normalize the Batch score as $1 - S_{batch}$ so that higher is always better.

- **Clustering Metrics (NMI, ARI)**: We perform Leiden (Traag et al., 2019) clustering on the embedding and compare the resulting clusters to the ground-truth biological labels using Normalized Mutual Information (NMI) and Adjusted Rand Index (ARI) (Vinh et al., 2010).

- **Batch Correction (1 - F1)**: We train a k-NN classifier to predict the batch identifier from the corrected embeddings. The F1 score of this classifier measures the remaining batch signal. We report $1 - F1$, where a score of 1 means the batches are indistinguishable.

### C.2  MEAN AVERAGE PRECISION (MAP) SPECIFICS

The mAP metric evaluates the model's ability to retrieve biologically similar samples. For each sample (query), we rank all other relevant samples by cosine similarity.

- **Positive Samples**: Other replicates of the same compound (matching InChIKey).

- **Negative Samples ('mAP control')**: All negative control wells from the same plate as the query.

- **Negative Samples ('mAP no rep')**: All wells from the same plate treated with different compounds.

The Average Precision (AP) is calculated for each query, and the mAP is the mean AP across all queries. The full mathematical formulation is provided in (Arevalo et al., 2024).

