# OpenReview forum: "CellPainTR: Generalizable Representation Learning for Cross-Dataset Cell Painting Analysis"
_ICLR.cc/2026/Conference — Submitted to ICLR 2026_

### Official Review · Reviewer_sEwB · 2025-10-14

**Soundness:** 1
**Presentation:** 1
**Contribution:** 1
**Rating:** 0
**Confidence:** 5

**Summary:**

This paper proposes a transformer model designed specifically to represent and batch-correct the manually engineered features coming from a specific tool known as CellProfiler (which extracts thousands of "interpretable" features from cellular images). The authors design a complex model called CellPainTR to create new representations that post-process the CellProfiler feature vectors. Results on JUMP-CP and other datasets suggest that their model modestly can improve over much simpler non-transformer-based batch-correction methods on the raw CellProfiler features.

**Strengths:**

1. The ablation approach is a good start to understanding the necessary features of their model on the tradeoff between their SSL training step versus their supervised contrastive learning training step.

2. The proxy adaptation strategy in using a specialized source token for adapting to a new domain seems potentially promising.

3. There is some awareness of the limitations of this work in lines 471-472: "evaluation against a broader array of deep-learning-based methods would be valuable" -- not only would it be valuable, but it would be required for publication at a general-purpose conference on the topic of learning representations.

**Weaknesses:**

There are a multitude of weaknesses in this work which make it inappropriate for ICLR and ML conferences in general, in its current form.

1. The entire approach presumes that CellProfiler features are an appropriate orientation for cutting-edge ML research, despite a large amount of related work having indicated that representations from deep self-supervised image models can significantly outperform CellProfiler features. Deep vision models such as MAEs extract richer biological signal and in enabling more robust batch-correction [see, e.g., A,B,C]. There is no mention of such image representation learning models nor any comparison to them. Indeed, if the authors had attempted to apply their representation-learning regime to batch-correct such representations, and shown significant improvements over the state-of-the-art method (typical variation normalization), the results could perhaps have been more interesting and generalizable.

2. Even given the context of these CellProfiler features (already problematic, as indicated above) there is no substantive comparison to other ML representation learning methods. They evaluate a transformer against old methods like Combat, Harmony, and Sphering (furthermore, this is somewhat difficult to decipher as there is no direct discussion of what the method labeled as "Baseline" actually is, or what "Sphering" is, within their paper). The authors could have tried a much more simple and comparable ML baselines to their transformer: simply fit and transform PCA on the CellProfiler embeddings, or train some kind of autoencoder. Most importantly, they should have applied methods like Typical Variation Normalization (TVN), a critical and SOTA step in batch correction in biological data processing from scRNAseq to microscopy [see, e.g., D, E]. As well, it is concerning why the Sphering baseline (again completely undiscussed) was only included in Table 1 and not Table 2.

3. The language in the paper is often grandiose and exaggerated throughout ("drastically outperforms", "striking", "dramatically", "not only / not just X [...] but Y [...]", etc), suggesting the overuse of LLMs. For example, consider this section: "4.3 THE CRITICAL TEST: OUT-OF-DISTRIBUTION GENERALIZATION The true measure of a foundational model is not just its performance on held-out data from the same distribution, but its ability to generalize to entirely new, unseen datasets." It is completely inappropriate to claim that their method is a foundational model, given that it is simply post-processing very specific manually extracted features from another software (not even processing the images directly). Also, see lines 375-377 where a strange quotation mark is included at the end, suggesting potential copy-pasting an explanation outputted from an LLM that rationalized the observed performance metrics.

4. There is no analysis of the internal learned representations from their model. In lines 203-204 they describe the feature-specific representations learned by CellPainTR. Surely a simple analysis or sanity check of these representations would be appropriate and revealing. For example, one might expect that the features corresponding to CellProfiler AreaShape would cluster together, as well as those corresponding to Intensity. No such analysis of the inner representations learned in their model is provided.


References:

A - Kraus, Oren, et al. "RxRx3-core: benchmarking drug-target interactions in high-content microscopy." arXiv preprint arXiv:2503.20158 (2025). LMRL Workshop at ICLR 2025.

B - Kenyon-Dean, Kian, et al. "Vitally consistent: Scaling biological representation learning for cell microscopy." arXiv preprint arXiv:2411.02572 (2024). ICML 2025.

C - Kim, Vladislav, et al. "Self-supervision advances morphological profiling by unlocking powerful image representations." Scientific Reports 15.1 (2025): 4876. Nature 2025.

D - Celik, Safiye, et al. "Building, benchmarking, and exploring perturbative maps of transcriptional and morphological data." PLOS Computational Biology 20.10 (2024): e1012463.

E - Ando, D. Michael, Cory Y. McLean, and Marc Berndl. "Improving phenotypic measurements in high-content imaging screens." BioRxiv (2017): 161422.

**Questions:**

N/A

---

### Official Review · Reviewer_JehU · 2025-10-18

**Soundness:** 2
**Presentation:** 3
**Contribution:** 2
**Rating:** 2
**Confidence:** 4

**Summary:**

This paper introduces CellPainTR, a multi-stage, multi-component framework designed to improve cross-source generalization in Cell Painting image analysis. The model leverages components such as feature context embedding, source context tokens, and linear adapters to capture source-specific and perturbation-related signals. It employs a self-supervised pretraining stage, intra-source contrastive training, and inter-source alignment to learn robust representations across datasets. The goal is to address domain shifts and enhance downstream performance in out-of-distribution (OOD) settings.

**Strengths:**

The paper addresses an important and challenging problem, cross-source generalization in Cell Painting data, with significant practical relevance. It proposes a comprehensive framework that integrates multiple components, including feature embeddings, source context tokens, and linear adapters, within a multi-stage training pipeline to jointly address representation learning and domain adaptation. The work is well-motivated by the limitations of existing batch correction and embedding methods.

**Weaknesses:**

While this paper proposes a novel model architecture for integrating Cell Painting data, several significant issues remain unresolved:

1. **Unclear Component Contributions**: The motivation and individual roles of each module (e.g., linear adapter, feature context embedding, source context token) are insufficiently explained. It is unclear how these components contribute to generalizability. For example, for the linear adapter, which seems to be another MLP layer with output dimension $C$. The benefit is unclear of the linear adapter is unclear, given that Hyena already provides a learnable weight matrix at the sequence level.

2. **Ambiguous Feature Representation & Feature Context Embedding**
The paper does not clearly describe how cell features, $C$, are obtained. Whether they are raw Cell Painting images or precomputed embeddings (e.g., CellProfiler-derived features).  Similarly, the definition of $L$ in the feature context embedding is unclear. If $L$ represents the number of images per perturbation, the connection to positional encoding should be discussed. If $L$ corresponds to channel dimensions, prior works leveraging channel-wise token embeddings [1,5] should be cited and discussed.

3. **OOD Generalization Strategy**
The idea of a source context token $S_k$ is interesting, but its effectiveness for out-of-distribution generalization is unclear. Given that the unseen source tokens are not encountered during training, how this strategy would help generalizability is questionable.

4. **Incomplete Methodological Details**
The SSL pretraining loss closely resembles existing approaches [1] but is not properly cited. Furthermore, for supervised contrastive learning, the author did not mention how positive and negative samples are selected during contrastive training and how these choices influence model performance.

5. **Multi-stage Training Process**
The details of Stage 3 (inter-source training) are insufficient. It is unclear how contrastive pairs are selected across sources (e.g., same perturbation across different sources) and how the model handles unseen source tokens. In addition, the author did not fully discuss the necessity and benefits of the multi-stage training pipeline. For example, why not train directly from Stage 2?

6. **Limited Baseline Comparisons**
The paper only compares against traditional batch correction methods and its own variants, e.g., CellPainTR variants. However, there is a line of related embedding methods, such as self-supervised learning [1,2] or cross-modal contrastive learning methods [3,4,5], and weakly supervised methods [6], that should be considered as baseline comparisons. Similarly, the paper claims that Hyena is computationally more efficient than self-attention, but no runtime or comparison to other transformer-based baselines [1,2] is provided.

7. **Insufficient Ablation Studies**
Despite the proposed multi-component architecture and multi-stage training, the paper lacks ablation studies examining how individual modules, feature types, training stages, or loss coefficients affect the model performance.



[1] Masked Autoencoders for Microscopy are Scalable Learners of Cellular Biology Kraus et al CVPR 2024

[2] Unbiased single-cell morphology with self-supervised vision transformers Doron et al bioRxiv

[3] CLOOME: contrastive learning unlocks bioimaging databases for queries with chemical structures Sanchez-Fernandez et al Nat Com. 2023

[4] How Molecules Impact Cells: Unlocking Contrastive PhenoMolecular Retrieval Fradkin et al NeurIPS 2024

[5] CellCLIP – Learning Perturbation Effects in Cell Painting via Text-Guided Contrastive Learning Lu et al NeurIPS 2025

[6] Learning representations for image-based profiling of perturbations Moshkov et al Nat Com. 2024

**Questions:**

See weakness

---

### Official Review · Reviewer_8pMc · 2025-10-30

**Soundness:** 3
**Presentation:** 3
**Contribution:** 2
**Rating:** 4
**Confidence:** 3

**Summary:**

This work introduces a transformer-based architecture for integrating cellular morphology from Cell Painting assays, with the primary objective of mitigating batch effects arising from laboratory and plate variation. The model operates on a large set of hand-crafted morphological features provided by the Cell Profiler software, a popular tool for the analysis of Hight Content Screening data. A source token is appended to the sequence; each real-valued feature is linearly embedded and combined with a token encoding the feature’s identity. The resulting sequence is then passed through a transformer in which the standard self-attention mechanism is replaced by a HYENA operator.

The model is evaluated against several baselines on the JUMP-CP dataset

**Strengths:**

1. The paper addresses an important problem in High Content Screening, and Cell Painting in particular. Indeed, it is often difficult to integrate data due to batch effects that heavily impact phenotypic readouts.
2. The paper is overall well written and relatively easy to understand. The method is overall well designed and architectural choices are mostly well justified. The chosen datasets and evaluation metrics are appropriate for the research question.

**Weaknesses:**

1. The authors motivate the use of Cell Profiler features for their biological interpretability but they do not show any use case for this interpretability feature which raises the question of why using these features in the first place. At the same time, since now the field has moved to SSL features for this kind of assays (e.g. https://doi.org/10.1038/s41598-025-88825-4 ), it would be interesting to compare the biological and batch metrics for this kind of features.
2. The curriculum is not well motivated and the setup needs clarification: How many epochs does each stage last? Why is simultaneous optimization of all three losses not considered a viable alternative?
3. In the ablation study, the improvement with respect to batch and biological metrics over the different stages of training is unclear.
4. In the experiments, the term “biological signal” is not defined. What exactly does it denote in this context, e.g., mechanisms of action (MoA) ?
5. The manuscript never clearly defines the “baseline” method or the “sphering” procedure against which they compare their method.
6. The adaptation of Transformers for tabular data is not as novel as presented in the paper. Previous published work such as (https://dl.acm.org/doi/10.1145/3704728) have already adapted the transformer for tabular data but no literature on this topic is presented in the study.
7. The source token is not properly discussed; it is not clear what metadata is used to define it.
8. Methods using contrastive learning usually perform best when increasing the batch size (up to 4096 in methods like MoCov3; https://arxiv.org/abs/2104.02057) , however the authors used a batch size of only 32 / 64. The inconsistent changes in metrics make the use of the contrastive framework questionable in this context.
9. The authors claim to train a foundation model which is broadly applicable without retraining. However, they do not explain how they found the most similar proxy label from the training data

**Questions:**

1. How are the scores normalized in table 1 ?
2. Why do Harmony and Combat perform very poorly on table 2 if they are re-fit ?
3. Can you define more clearly InChIKey ?
4. Why introducing groups of features in training stage 1 ? It is not clear how these groups are used since the masking is applied uniformly
5. Why do Harmony and ComBat perform as badly as they perform in table 2 (way worse than in table 1), if they have been refit to the data ?

---

### Official Review · Reviewer_7mQz · 2025-11-01

**Soundness:** 3
**Presentation:** 3
**Contribution:** 2
**Rating:** 4
**Confidence:** 3

**Summary:**

This paper presents CellPainTR, a Hyena-based Transformer that learns batch-robust representations from Cell Painting features. The model embeds continuous morphological features with a linear adaptor, encodes feature identity, and conditions on a source token to capture data provenance. Training follows three stages: self-supervised masked feature reconstruction, intra-source supervised contrastive learning, and inter-source supervised contrastive learning to enforce source-invariant embeddings. It achieves the best overall balance among baselines and outperforms retrained methods even without fine-tuning under severe feature mismatch.

**Strengths:**

- This paper performs well in terms of novel architecture and efficiency. The model replaces standard self-attention with Bidirectional Hyena operators, enabling near-linear complexity over thousands of features and making Transformer-style modeling feasible for Cell Painting profiles.
- The model uses learnable source context tokens to explicitly model data provenance, providing a principled way to capture and mitigate source-specific batch effects while preserving biological signal.
- The three-stage curriculum progressively builds invariance while preserving biology.
- The OOD result shows meaningful generalization without fine-tuning.

**Weaknesses:**

- The baseline comparisons are narrow, as they exclude recent deep learning and self-supervised representation methods.
-  The exclusive focus on engineered feature space may limit performance compared with end-to-end image models, and the paper does not quantify the trade-off or explore joint feature–image training.
- The proxy-token strategy may be sensitive to metadata quality, which could reduce reliability in real deployments.

**Questions:**

Please refer to my weakness section.

---

### Meta-Review · Area_Chair_RhFY · 2025-12-19

**Summary:**

This paper looks at the problem of integrating data from multiple heterogeneous cellular measurements and learning general representations for cell morphology. The paper has received very negative reviews and the authors did not submit a rebuttal. I believe they possibly meant to retract the paper, it’s clear that the reviewers would not have changed their minds. The reviewers raised many limitations, from poor baselines, excessive feature engineering, biological interpretability, unclear methodology with many stages whose contributions are unclear and more.

Given the initial scores and lack of a rebuttal, this is a clear rejection.

**Reviewer Concerns:**

The reviewers found many isses with the paper, all of which remain outstanding given the lack of a rebuttal.

**Reviewer Scores:**

The reviewers would not have changed their score.

---

### Decision · Program_Chairs · 2026-01-26

Reject